# Activation of Host Cellular Signaling and Mechanism of Enterovirus 71 Viral Proteins Associated with Hand, Foot and Mouth Disease

**DOI:** 10.3390/v14102190

**Published:** 2022-10-04

**Authors:** Subrat Kumar Swain, Subhasmita Panda, Basanta Pravas Sahu, Rachita Sarangi

**Affiliations:** 1Department of Paediatrics, Institute of Medical Sciences and SUM Hospital, Siksha ‘O’ Anusandhan (Deemed to Be) University, Bhubaneswar 75003, India; 2School of Biological Sciences, University of Hong Kong, Pok Fu Lam Road, Hongkong, China

**Keywords:** Enterovirus, viral proteins, signaling pathways, host-pathogen interaction

## Abstract

Enteroviruses are members of the *Picornaviridae* family consisting of human enterovirus groups A, B, C, and D as well as nonhuman enteroviruses. Human enterovirus type 71 (EV71) has emerged as a major cause of viral encephalitis, known as hand, foot, and mouth disease (HFMD), in children worldwide, especially in the Asia-Pacific region. EV71 and coxsackievirus A16 are the two viruses responsible for HFMD which are members of group A enteroviruses. The identified EV71 receptors provide useful information for understanding viral replication and tissue tropism. Host factors interact with the internal ribosome entry site (IRES) of EV71 to regulate viral translation. However, the specific molecular features of the respective viral genome that determine virulence remain unclear. Although a vaccine is currently approved, there is no effective therapy for treating EV71-infected patients. Therefore, understanding the host-pathogen interaction could provide knowledge in viral pathogenesis and further benefits to anti-viral therapy development. The aim of this study was to investigate the latest findings about the interaction of viral ligands with the host receptors as well as the activation of immunerelated signaling pathways for innate immunity and the involvement of different cytokines and chemokines during host-pathogen interaction. The study also examined the roles of viral proteins, mainly 2A and 3C protease, interferons production and their inhibitory effects.

## 1. Introduction

Enterovirus71 (EV71) is a sense, single-stranded, non-enveloped RNA virus belonging to the *Enterovirus* genus of the family *Picornaviridae,* which is the main causative agent of hand, foot and mouth disease (HFMD) that can have serious health complications, such as neurological and cardiovascular complexities, and commonly affects infants and children [1]. This disease is manifested by a wide range of symptoms such as fever, rash, diminished appetite, and mouth ulcers followed by opportunistic critical clinical conditions, such as neurological dysfunction, cardio-respiratory collapse, or evenmortality [2]. The 7.4 kb long genome of EV71 is a single positive-strand RNA and has an open reading frame (ORF) that codes for a polypeptide of 2194 amino acids followed by the 3’ non-translated region with a poly-adenylated (poly-A) tail. In the case of the EV71 genome, an internal ribosomal entry site (IRES) has been observed with the facility of cap-independent translation in the viral protein. The ORF has a single wide polyprotein witharound 2100 amino acids and separated into three regions (P1–P3). A variety of processing activities in the polyprotein lead to the cleavage of the polyprotein and viral non-structural and structural proteins. The P1 region constitutes the capsid virus and encodes the four structural proteins including VP1, VP2, VP3, and VP4 (VP1–VP4). The virus replication’s most direct results are non-structural P2 proteins (3Dpol, 3CDpro, 3Cpro, 3A, 3AB, and 3B) [3]. The capsid of the EV71 is developed through the protomeric components of 60 units and involves the four structural protein polypeptides VP1-VP4. These are encoded in the genome P1 area. P3 and P2 encode seven nonstructural proteins (3A–3D and 2A–2C), followed by poly-A residues at 3’UTR. Among these, it has been demonstrated that the viral protease 3C is implicated in numerous pathogenic processes of EV71. The connection site Gln-Gly of P2-P3 can be cleaved by EV71 3C proteases [4]. 

Host-pathogen interaction provides scope for the development of host receptors based on the identification of specific routes for the interaction. It is important to understand that the host-viral relationship is responsible for promoting the viral protein to facilitate the condition of generating a host factor to interact with the viral protein. The process of infection is associated with the condition of the host and its immune response [5]. When an EV71 infection occurs, inflammation is known to play a critical role. This is always characterized by an infiltration of inflammatory cells, a release of pro-inflammatory cytokines and chemokines, edema, and vascular leakage [6]. It is widely recognizedthat EV71 infection can result in complex inflammation insites accompanied by immune evasion, multiple immune cell responses, and proinflammatory cytokine release [7,8]. It has been reported that many cellular signaling pathways are involved in EV71 replication and inflammatory pathogenesis [9,10,11]. In this review, we mainly focus on the major cellular signaling pathways involved in EV71-induced antiviral innate immunity and inflammatory responses. 

## 2. Molecular Mechanisms Associated with Host-Pathogen Interaction in EV71

The molecular mechanisms involved in the host-pathogen receptor are associated with the identification of the cell surface receptor development for entrapment of the virus. It is important to understand the five different molecules on the cell surface of the host, which mediate the host-pathogen interaction by the involvement of possible cell surface receptors. The five individual molecules functioning as cell surface receptors are as follows: Scavenger receptor B2 (SCARB2), heparan sulfate, Sialylated glycan, P-selectin glycoprotein ligand-1 (PSGL-1), and annexin II [12]. The SCARB2 protein provides the cell-surface receptor for the development of binding sites, which is evidence for the critical role of receptor development in host-pathogen interactions. Moreover, EV71 is involved in the endocytic pathway duringthe progress of infection (Figure 1). 

Clathrin-mediated endocytosis (CME) is the principal endocytic mechanism in mammalian cells for the uptake of surface receptors (i.e., cargo) and their associated ligands. It also controls cell-cell and cell-substrate interaction, intracellular signaling, and cellular homeostasis. Six PRMPs in the C-terminal cytoplasmic tail region of SCARB2 allow the SH3 domain of endophilin to recognize SCARB2 and initiate endocytosis [13]. Despite being extensively expressed, endophilin-A2 is necessary for EV71 invasion in Caco-2 cells. The virus enters RD and NIH3T3 cells through a clathrin-dependent pathway, but Jurkat and L-PSGL-1 cells through caveolae [14]. One study using endocytosis inhibitors demonstrated that altering clathrin and dynamin did not suppress, but rather increased, EV71 infection in A549 cells, suggesting an unknown dynamin-independent endocytic mechanism [15]. 

Chen et al., 2019 and Renard et al., 2015 found that endophilin A2 correlates with membrane scission during clathrin-independent endocytosis. C-SH3 domain interacts with proline rich motifs (PRMs) in active cargo receptors to recruit adaptors and cargo [16,17].

The viral capsid is an important part of the molecular mechanism for host-pathogen interaction to promote the development of the natural lipids that help to uncoat the viral particle to release the genome inside the host. The mechanism of action of the viral capsid is developed through conformational changes in the capsid followed by the removal of lipids thatenablesthe release of VP1 and VP4 N-termini to support uncoating of the viral genome. However, in the case of human EV71 and Coxsackievirus A16, two types of membrane proteins such as SCARB2 and PSGL-1 provide the conditions for the development of the receptors. The important step of uncoating the viral genome occurs in the endosome area, followed by the acidification to support the host-related interaction. However, it is important to understand that the acidic environment (pH = 4) is associated with the development of EV71-related host-pathogen interaction by promoting the releaseof the viral genome [18].

The mechanism and molecular context of the host-pathogen interaction indicate the concept for how viral replication occurs inside the host cell. The immune response of the host can be determined through the evaluation of the replication. In the process ofviral protein development, the host translation system translates the viral genome inside the host cytoplasm by the action of FUBP1 followed by an internal ribosome entry-transacting factor [19]. The internal changes that occur due to the outbreak of the viral protein lead to the suppression of antiviral and cap-dependent transcription by the presence of different signaling components such as PI4KB and immune responses, retinoic acid induced gene-I (RIG-I) as well as mitochondrial antiviral signaling protein (MAVS). Theseinteractions develop through the involvement of SiRNA libraries of endocytosis, serine or threonine kinase as well as the genes of membrane trafficking.

On the one hand, it is necessaryto understand that post-transcriptional gene expression regulation is associated with the development of microRNA, which identifies the host and viral particle complexity during attachment. On the other hand, few genetic expressions play an inhibitory role during viral replication. A post-transcriptional factor named miR-23b enables the down-regulation ofinteractions. Meanwhile, 3’UTR of EV71 is a conserved sequence that is regulated by miR-23b. Moreover, the effectiveness of the virus replication within the host cell is associated with the up-regulation of the hsa-miR-494-3p level, which regulates the PI3K/Akt signaling pathway [20]. Furthermore, hsa-miR-141 expression targetstranslation initiation factor eIF4E based on the cap-dependent and cap-independent translation within the host system [21]. In this case, host miRNA is associated with the development of potential interactions to inhibit the action ofviral propagation. Moreover, the gene sequence hsa-miR-548 is responsible for the development of host antiviral responses by targeting the IFN-λ1 factor within the host organism.

## 3. Immune Cells Involvement during Evasion Process

Host-pathogen interaction is associated with the activation of immunogenic responses inside the body of the host organism. However, the invasion of aforeign particle or pathogen intothe host organism activates conflicts between the innate immune system and adaptive immune system [22]. However, it is important to understand that the immunogenic defense mechanism of the host organism is involved inactivating the adaptive immune system throughout the functional condition of the innate immunity [23]. Innate immunity is responsible for a range of physiological responses thatfacilitate pathogenic evasion inside the host organism through various immunogenic cells such as macrophages, neutrophils, dendritic cells, and natural killer cells. Moreover, it can be observed that the innate immune system of the pathogen enables the identification of pathogen-associated molecular patterns (PAMPs) through the development of pathogen-recognition receptors (PRR) on the cellular organelles and membrane to identify the foreign particle.

### 3.1. Study of Animal Pathogenesis Using EV71

EV71 infection is normally mild and self-limiting but it can lead to CNS infection, aseptic meningitis, brain stem encephalitis, and severe flaccid paralysis. Fatal infections are associated with severe neurological consequences, pulmonary edema, and haemorrhage [24]. In cynomolgus monkey, it causes CNS lesions [25,26]. Expression of the EV71antigen in infected rhesus monkey organs was used to studythelink between EV71 tissue replication and disease. The CNS, lungs, and bronchial tubes expressed EV71 antigen highly [27]. After infection with a mouse-adapted viral strain, the murine model showed gradual immobility, ruffled fur, humped posture, and death [28]. Mouse L929 cells transformed with hSCARB2 are sensitive to all human enterovirus species A strains [14,29], with enhanced virion binding, internalization, and uncoating. PSGL-1transfected mouse L929 cells allow EV71 invasion, replication, and cytopathic effects [30]. The antigens were detected in the animal brain stem, spinal cord, skeletal muscle, and lungs. Humans, monkeys, and transgenic mice show similar EV71 neurotropism [31,32].

### 3.2. Apoptosis

As soon as EV71 enters the host cells, cap-independent translation of the viral RNA starts. The recruitment of the generated viral proteins triggers the replication of viral RNA. Apoptosis usually starts as the viral lineage is being replicated by the host or viral components [33]. According to recent studies, autophagy, particularly in the case of neuronal cells, may also be responsible for the cell death associated with the EV71 infection. Increasing evidence indicates that EV71 infection induces apoptosis in a range of cell lines, including HeLa, rhabdomyosarcoma (RD), Jurkat, SK-N-MC, glioblastoma SF268 ce, Vero, and the human microvascular endothelial cells [34,35,36,37]. Numerous investigations have demonstrated that the activation of caspases by the proteolytic activity of EV71 3C and 2A can result in apoptosis. EV71 also induces apoptosis in a variety of host cells via many apoptotic mechanisms. Caspase 8 activation and Bid cleavage are produced by EV71 infection in non-neural cells, whereas the mitochondrial route causes apoptosis in neural cells [38]. In addition, Chen et al., 2006 found that EV71-infected brain cells activated Cdk5. The increased FasL expression that has been observed in Jurkat T cells infected with virusmay help to explain the decrease in T cells [39]. Notably, it has been demonstrated that very early in the course of infection, EV71 infection activates the PI3K/AKT and MAPK/ERK signaling pathways. The stimulation of these pathways inhibits GSK3b activity and may delay host apoptosis [40].

Infection with EV71 has been found to change cellular signaling cascades such as MEK/ERK and PI3K/Akt in order to modify the cellular function and the virus life cycle [40,41,42]. The PI3K/Akt pathway activation may be implicated in the regulation of viral protein production and host cell apoptosis [41]. Activated ASK1 is a substrate for Akt phosphorylation, which is related toa reduction in stimulated ASK1 kinase activity. Akt inhibits ASK1 activity, resulting in the observed restricted JNK phosphorylation. EV71 infection of RD cells activatesthe pro-apoptotic protein Bax, as shown by a conformational shift and translocation from the cytosol to mitochondria, which coincideswith the release of Cytochrome C. However, the PI3K/Akt survival pathway in RD cells restrictsthe degree of EV71 induced JNK activation and JNK-mediated death via Ask1 phosphorylation [43] (Figure 2).

### 3.3. Autophagy

In the host defense against a variety of intracellular pathogens such as bacteria, parasites and viruses, autophagy probably plays a significant role. Double-membrane autophagosomes elevate in RD and neuroblastoma (SK-N-SH) cells that have been infected with the respective virus. The expression of the autophagosomal marker LC3 II and the viral proteins are positively correlated [34]. The mechanisms by which EV71 induces autophagic processes should be understood in order to design effective treatment plans for EV71-related disorders. Toll-like receptor (TLR) activation or induced endoplasmic reticulum (ER) stress can cause autophagy [44]. Jheng et al. 2010 showed that EV71 infection causes eIF2a to become phosphorylated and leads to the overexpression of the ER-resident chaperone proteins BiP and calreticulin [45]. Since inactivated EV71 is devoid of these effects, ER stress might be brought on by viral replication rather than viral entrance or attachment that leads to development of autophagosomes.

### 3.4. Innate Immunity

The activation of the PRR pathway, which results in the generation of IFNs, is typically linked to the host innate immune response. TLRs and DExD/H box helicases such asRIG-I and melanoma differentiation associated gene (MDA5) are among the PRRs [46]. Many cells express TLR, and notably, TLR3, 7, and 8 are capable of detecting RNA inside cells. Therefore, these TLRs are crucial for detecting viral incursions. MDA5 and RIG-I are homogenous IFN-inducible proteins that are triggered by RNA viruses. The IFNs that are created have antiviral properties because they can attach to receptors on the small cell or on nearby cells. According to onestudy, EV71 3C can connect with RIG-I. This interferes with IPS-I recruitment, which in turn prevents IRF3 from being activated [47].

### 3.5. Acquired Immunity

The immune system is triggered by viral infection to eliminate the invasive pathogens. Age is a crucial risk factor for this disease. Children under the age of five with undeveloped immune systems are particularly vulnerable to EV71 infection [48,49]. Age-related increases in EV71 antibody titers have been demonstrated by retrospective serological research [50]. Thus, this might provide an explanation for why young toddlers are vulnerable. Eighty percent of individuals with hand, foot, and mouth disease whose antibody titers were examined tested positive for the antiEV71 antibodies within a day of developing their illness [51], but there is no connection between the severity of the illness and the antibody response [51,52]. Therefore, the fate of an EV71 infection may be significantly influenced by the cellular immune responses.

### 3.6. Immune Evasion

The process of immunogenic evasion is responsible for an increased probability that pathogens spread by escaping the immunogenic responses within the host organism. Host immune pressures may result in the development of immunological evasion caused by EV71 resulting in the spread of the virus inside the host cell.

## 4. Characterization of Coding and Non-Coding Regions of EV71

### 4.1. VP1

VP1 is a primary 297-amino acid capsid protein present at the most exterior portion. During EV71 infection, VP1 binds to the immunoglobulin–like receptors PSGL-1 and SCARB2. Receptor biding requires PSGL-1 binding by regulating VP1-244K exposure. When EV71 virions attach to the receptors, they undergo a two-step uncoating process, first creating an enlarged, altered “A-particle” that expels VP4 and exposes the N-terminus of VP1. Second, the A-particle capsid develops a breach in the endosomal membrane and opens a two-fold channel near the icosahedral axis to facilitate genome release. The crystal structure of the EV71 uncoating intermediate also revealsthat the VP1 N-terminal extensions (1-71residues) interact with the viral RNA. Furthermore, VP1 activates ER stress, and autophagy may enhance the overexpression of cell surface-exposed calreticulin (Ecto-CRT), a key mediator of primary phagocytosis. Heparan sulfate is used as an attachment receptor by EV71, and neither VP1-98 nor VP1-145 can modify heparan binding. Galectin-1, a soluble beta-galactoside binding lectin, may interact with VP1 via carbohydrate residues before being released and attached to another cell surface alongside the virus. Vimentin on the cell surface is an attachment receptor that binds to VP1 via the N-terminus to enhance the infection. The VP1 protein activates calmodulin dependent protein kinase II, which phosphorylates the N-terminal region of vimentin on serine 82 during infection. Vimentin phosphorylation and rearrangement may improve EV71 replication by playing structural roles in replication factor production.

### 4.2. VP2–VP3

The EV71 virus capsid proteins VP2 and VP3, which are important parts of the shell protein, are associated with the antigenicity of the virus. Thus, VP2 and VP3 may be potential candidates with structuressimilar to that of VP1, and VP2 (amino acids 142–146) contains a single, linear, non-neutralizing epitope, which is located in the E–F loop of the VP2 protein [53,54]. The structure of VP2, VP3 and VP1 is the same; they form an eight stranded anti-parallel b-barrel structure in the shape of a wedge that favors packing. The shape resembles a b-sandwich “jelly roll” fold. The linking loops and C-termini on the outer surface of the capsid are the key structural changes. The “puff” is the most noticeable surface loop in VP2 and the “knob” is the greatest protrusion on the surface in VP3. VP3 consists of 245 amino acids, among which the amino acids 59–67 of VP3 are more highly conserved between the subgenogroups, compared withVP1. There is a conserved conformational epitope on the “knob” region of VP3, which makes it an ideal target for a diagnostic or therapeutic mAb. 5H7, a therapeutic IgG antibody, was recently demonstrated to target a conformational epitope mapped to highly conserved amino acid position 74 of VP3.

### 4.3. VP4

VP4 consists of 69 amino acids, an extended conformation, and is present inside the virion. The VP4 package is embedded in the virus shell, is strongly linked to the virus score, and has an extended spatial conformation characteristic, which acts as a bridge between the inside and outside of the virus [53,55]. The spatial configuration of the virus changes when it connects to the receptors. The viral shell is eventually destroyed, followed by the release of viral genomic RNA into the cytoplasm, and the viral translation begins. The viral genomic RNA, known as mRNA, is the starting point for the polymeric protein. The VP4 N-terminal myristoylation signal (MGXXXS) plays an important role in EV71 replication. The VP4 genome is more conserved than the VP1, VP2 and VP3 genes; thus, studies on EV71 vaccine have focused on neutralizing epitopes of the VP4 protein [56].

### 4.4. 2A–2C and 3A–3D

The EV71 2A protease has cysteine protease activity and 150 amino acid residues, and isan enzyme that cleaves at its own N-terminus at the polyprotein junction between VP1 and 2A. By cleaving the elongation factor elF4G and facilitating EV71 replication, 2Apro suppresses host cap-dependent protein synthesis [57,58]. By cleaving mitochondrial antiviral protein and RIG-1 like receptor MDA5, the 2A protease can lower IFN-1 receptor protein levels and block interferon regulatory factor 3 (IRF3) signaling, allowing the virus to evade the immune response [59]. 2A inhibits IFN- γ signaling by lowering serine phosphorylation of the signal transducer and activator of transcription 1 (STAT 1) [59,60,61].

The EV71 2B protein, a tiny hydrophobic ion channel protein with 99 amino acid residues, may mediate a chloride-dependent rather than calcium-dependent channel to regulate viral replication [62]. The C-terminal portion of 2B (63–80 amino acids) is thought to be crucial for mitochondrial localization and causes cell death via interacting with and recruiting Bax, a proapoptotic protein, as well as triggering Bax conformational activation. A 14-amino acid hydrophilic domain in the N-terminus of 2B is required for Bax binding and subsequent activation (Figure 3) [63].

2C protein, which has 329 amino acid residues, is one of the most highly conserved nonstructural proteins. At the C-terminal region, it contains an ATPase domain, a zinc finger structure, and an alpha helix [64]. In vitro, the 2C ATPase, an RNA helicase that unwinds RNA helices in an ATP-dependent manner and an RNA chaperone independent of ATP, promotes EV71 RNA production. Through a PP1-docking motif, the N-terminus of 2C (1–125 amino acid) interacts with all isoforms of the protein phosphatase 1 (PP1) catalytic subunit, which is effective for EV71 2C-mediated suppression and NF-κB activation (Figure 3) [44].

### 4.5. 2A and 3C Proteases

2A protease is initially translated during the translation of the enterovirus polyprotein non-structural region (P2). It then self cleaves to split from the P2 and P1 areas. This is followed by the P3 region, which includes the second protease 3C, which is responsible for 8 out of 10 cleavages of the viral polyprotein [65]. Different enterovirus genotypes showed roughly 50% to 75% sequence similarity in 2A and 3C. The twisted β-barrels stack perpendicular to each other from the tertiary structure of 3Cpro.The domains contribute to the formation of a catalytic site, which includes histidine, aspartic acid and cysteine in 2Apro and histidine, glutamic acid and cysteine in 3Cpro [66]. The P1 location, which is mainly glycine, is critical for 2A protease. The P2 position is highly significant after P1, which is commonly identified by threonine and asparagine. P2 might be proline, alanine, or phenylalanine, and P4 is generally a location for leucine or threonine. P1 in the substrate sequence shows the greatest conservation for 3Cpro [67]. For P1, the present amino acid is glutamine or glutamate, whereas, for P2 it is glycine, asparagine, or serine. 3Cpro has been identified as a promising target for antiviral medicines because the enteroviral polyprotein that contains many cleavages sites unique for protease and plays an important role in virus maturation [68].

## 5. Immune Related Signaling Pathways Associated with EV71

### 5.1. MAPK Signaling Activated by EV71

Mitogen activated protein kinase (MAPK) belongs to a family of Serine/threonine protein kinase, which is highly conserved throughout eukaryotes and is involved in a variety of cellular functions such as inflammation, stress, cell growth, cell development, motility, differentiation, proliferation and death by phosphorylated transcription factors and enzymes [69]. Mitogen activated protein kinase kinasekinase (MAPKKK), mitogen activated protein kinase kinase (MAPKK), and mitogen activated protein kinase (MAPK) are all sequentially phosphorylated in order to transmit the MAPK signal. There are six sub-familiesof MAPK that havebeen identified in mammalian cells to date: JNK1/2/3, ERK 1/2, p38 MAPK (p38α/β/γ/δ), ERK 7/8, ERK3/4 and ERK5/big MAPK1 (BMK1) [70]. In human cells, EV71 infection inducesintrinsic apoptosis and p38-mediated proinflammatory cytokines. Leong et al., 2015 discovered that EV71 infection led to the activation of misshapen/Nck-interacting kinase (NIK)-related kinase (MINK), which in turn increased the phosphorylation of p38. TAK1 is a participant in MAPKKK [71]. To prevent cytokine synthesis brought on by an EV71 infection, 3Cpro can affect the TAK1 complex proteins. As a result, EV71 infection, on the one hand, causes an inflammatory reaction by activating MAPK pathways [72,73]. On the other hand, in order to engage with the inflammatory response, EV71 may block MAPK pathways (TAK1 signaling).

The JNK1/2 and p38 MAPK signaling pathways regulate proinflammatory cytokine release as well as EV71 replication [74,75]. However, there is lack of evidence with regard to activation of JNK1/2 and p38 MAPK in immature dendritic cells (iDCs) during the viral infection. JNK1, JNK2, and JNK3 are the three different genes that code for the mammalian JNKs, and studies have shown that these proteins are highly active in response to cytokines, UV radiation, growth factor deficiency, DNA-damaging chemicals, and viral infection [76,77]. While JNK3 is exclusively present in the brain and testis; JNK1 and JNK2 are expressed in the majority of cell types [78]. A prolonged, EV71 infection boosts the mRNA levels of MEK4, MEK7, JNK1/2 and improves JNK1/2 phosphorylation [79].

The term “p38 MAPK” refers to four distinct isoforms of the protein [80]. Dual MAP2Ks (such as MEK3 and MEK6) and a number of MAP3Ks, including MTK1, MLK2/MST, MLK3, ASK1, and TAK1, have been observed to activate p38 MAPK kinases [70,81].

When EV71 infects iDC, it can activate the JNK1/2 and p38 MAPK signaling pathway cascades, phosphorylate downstream molecules such asc-Jun and c-Fos, and promote the release of proinflammatory cytokines. Proinflammatory cytokines such asIL-6, TNF, and IFN are frequently activated by oxidative stress, cytokines, and viral infection. These elements play an important role in inducing host cell damage, chronic inflammation, and other immune responses. In response to EV71 infection, DCs may release a variety of cytokines [82].

### 5.2. EV71 Induces Phosphatidylinositol 3-Kinase (PI3K) Signaling

Numerous Akt target proteins are phosphorylated as a result of Akt pathway activation, mediating a variety of cellular functions. Apoptosis signal-regulating kinase 1 [82], caspase-9, and the transcription factors Forkhead BAD, and NF-B are among the targets of Akt that have been linked to the regulation of cell survival [83]. The Akt pathway should typically activate a number of its downstream substrates. Inhibition of the PI3K/Akt pathway enhancesJNK phosphorylation and the JNK-mediated apoptosis during infection. Moreover, the PI3K/Akt pathway phosphorylatesapoptosis signal-regulating kinase 1 (ASK1) and negatively regulates ASK1 activity. Knockdown of ASK1 significantly decreased JNK phosphorylation, which inhibited the ASK1-mediated JNK activation. Collectively, these data reveal that activation of the PI3K/Akt pathway limits JNK-mediated apoptosis by phosphorylation and inactivating ASK1 during EV71 infection [43].

Though the activation of the downstream protein kinase AKt, PI3K signaling is well established for controlling cellular growth and proliferation and for playing a significant role in activating the inflammatory responses. According to some theories, EV71 stimulates the PI3K/Akt pathway, which in turn controls the transcription of pro-inflammatory cytokines and may be released in response to PI3K/Akt activation in human RD cells. Akt is phosphorylated by EV71 in a manner that is PI3K dependent. The PI3K/Akt pathway can be activated by EV71 infection, further influencing the inflammatory response.

### 5.3. EV71 Activates Calcium (Ca2+)-Dependent Signaling

Calpains are a family of calcium (Ca2+) dependent cysteine proteases. Both calpain 1(µ-calpain) and calpain 2 (m-calpain), which are ubiquitous isoforms localized in the cytosol and mitochondrion respectively, are activated invitro by micromolar and millimolar amounts of Ca2+. Due to the fact that a large number of calpain substrates are connected to the pro-apoptotic state, their activity has been linked to apoptosis [84,85,86]. Additionally, Ca2+ is an important regulator of cell survival; persistent Ca2+ elevation inside cells may activate calpains after exposure to some anti-cancer drugs, leading to apoptotic cell death.

Ca2+ is a ubiquitous second messenger that controls a variety of processes in eukaryotic cells. Infected cells with EV71 have been shown to have higher levels of mitochondrial Ca2+.Calpain activation through Ca2+ flow is crucial for triggering an apoptosis inducing factor (AIF), caspase-independent, apoptotic pathway during EV71 infection. In EV71 infected cells, the administration of ruthenium red, a mitochondrial Ca2+ influx inhibitor, dramatically inhibits calpain activation and AIF cleavage. Ca2+ homeostasis requires calmodulin-dependent protein kinase II (CaMKII). CaMKII can be activated by the EV71 VP1 protein. If activated, it phosphorylates serine 82 in the N-terminal domain of vimentin, which subsequently functions structurally in viral propagation. By triggering Ca2+ dependent signaling, EV71 infection promotes both virus replication and cell death.

### 5.4. EV71 Encoded Proteases Inhibit MAVS-Mediated Antiviral Signaling

The innate immune response is triggered as the initial line of defense against viral invasion when virus infects host cells. Host PRR detect PAMPs, which causes the release of type-1 interferon and proinflammatory cytokines that activate host adaptive immunity and generate an antiviral state in the host cells. Generally, there are three phases of anti-viral innate immunity: (i) the initiation phase whenPRR binds to viral RNA and recruits particular signaling adaptor molecules; (ii) the signal transduction phase in which adaptor molecules transmit signalsto activate IKK-related kinase that facilitate transcription factors such as interferon regulatory factor 3 (IRF3) and nuclear factor B (NF-B); and (iii) the effector phase when the Type 1 IFNcause the synthesis of interferon stimulated genes by activating the signal transducer and activator of the transcription (STAT) pathway. Membrane-bound TLRs and cytoplasmic sensors, such as RIG-I and MDA-5, are used to detect RNA viruses. RIG-I and MDA-5 both include RNA helicase domains and use the mitochondrial anti-viral signaling protein to transduce signals.

### 5.5. EV71 Infection-Associated IFN Signaling

Interferons are essential for limiting viral replication and dissemination in mammalian responses to viral infections [87]. Based on how they interact with their receptors, IFNs are divided into three types: type I, type II, and type III. Upon viral infection, type I IFNs are generated which then activate a number of antiviral effectors to build a defense network against viral replication [88]. It has been determined that type I IFN subtypes IFN-4, IFN-6, IFN-14, and IFN-16 suppress the replication of EV71. In particular, IFN-14 preventsvirus replication 20 times better than the traditional IFN-2a [89]. It is indeed significant to observe that EV71 inhibits IFN-mediated phosphorylation of STAT1, STAT2, JAK1 and Tyk2 via reductions inIFNAR1 through EV71 2Apro [90]. Another study found that IFN-induced IRF1 transactivation is blocked by EV71. However, there was no change in the expression of IFN receptors. A lack of type I and type II IFN receptors makes mice more likely to die after contracting EV71 [91]. It has also been revealed that a number of EV71 infection targets, including RIG-1, MDA5, MAVS, TLR9, TLR7 and miR-146a, interact with type I IFN responses. As a result, IFN-mediated signaling pathways play a crucial part in host innate immunity to EV71.

### 5.6. EV71 Interacts with IRF Signaling

Type I IFNs and IFN-inducible genes are crucial components of the immune system’s response to pathogen-derived danger signals [92]. It has been shown that EV71 3Cpro inhibits RIG-Imediated IRF3 activation and IFN-αproduction. Similar to this, EV71 deactivates IRF3 and severely reduces gene expression induced by IFN [93]. On the other hand, another study found that following EV71 infection ofHT-29 cells, IRF3 activated and translocated into the nucleus. In addition, it has been suggested that IRF7 rather than IRF3 mediates cleavage of EV71encoded 3Cpro. These contradictions need to be explained [94].

### 5.7. EV71 Triggers NF-κB Signaling

A protein complex called NF-κB regulates the transcription of several genes involved in immunological responses, cell proliferation, differentiation etc. Additionally, through promoting the expression of type I IFN, proinflammatory cytokines, NF-κB contributes significantly to the host antiviral response [95,96]. However, it has been noted that the EV71 2Cpro protein inhibits NF-κB activation and promotes virus replication. IKK and IBs kinase domains are directly contacted by the 2Cpro protein, which inhibits their phosphorylation. As a result, viral replication is boosted, and NF-κB activation decreases [97]. Evidence suggests that NF-κB signaling is necessary for respective virus replication and the inflammatory reactions [98]. As a result, NF-κBsignaling is crucial in the inflammatory response that EV71 causes and offers a possible antiviral approach (Figure 4).

### 5.8. EV71 Interacts with TLR Signaling

Recent findings have provided strong evidence of activation of a particular TLR that mediates the EV induced innate immune response [10,11]. There are 13 identified mammalian TLR which are expressed on the cell surface or in the endoplasmic reticulum. TLR1, 2, 4, 5 and 6 have been found on cell membranes and TLR3, 7, 8, 9, 10, 11, 12 and 13 have been found in endosomal compartments. Lipopolysaccharide, lipoprotein and flagellin are recognized by the TLR4/MD2 (myeloid differentiation) complex, TLR1/6, TLR2 and TLR5 to activate NF-κB and cause the generation of Type-I interferon. TLR7/8 recognizes the single stranded RNA of RNA viruses [99]. TLR3 recognizes the viral double-stranded RNA and recruits TRIF to induce TRAF3 and activate the TBK1/IKKƐ complex [100]. It can phosphorylate IRF7 or IRF3 to stimulate the production of IFN-I after activating TBK1/IKK. NF-κB is dimerized and enters the nucleus after activating the TAK1/TAB2/TAB3/TAB1 complex, which results in the production of proinflammatory cytokines [101].

Dendritic cells (DCs) are activated and matured by EV71 when TLR4 stimulates IB breakdown and NF-κB activation. According to Song et al., 2018, enhanced EV71 replication brought on by autophagy in 16HBE cells encourages endosome breakdown and suppresses the type I IFN response mediated by TLR7 [102,103]. The ESCRT-0 complex and the sorting of membrane protein in endosomes are controlled by hepatocyte growth factor-regulated tyrosine kinase substrates (HRS). During EV71 infection, HRS promotes the production of proinflammatory cytokines and interferons through the TLR7/NF-κB/p38 and TLR7/NF-B/IRF3 signaling pathways, which triggers inflammatory immune responses [104]. In contrast, TLR2, TLR7 and TLR8 mRNA were found to be upregulated inEV71infection human primary monocytes-derived macrophages at different time points and opposed to TLR3, TLR4, TLR6, TLR9 and TLR10. This finding implies that EV71 infection interacts with TLR signaling [105].

## 6. Preventive and Therapeutic Strategiesand Control of EV71 (Pathological Perspective)

Infections through the EV71 outbreak havebeen controlledby different therapeutic and preventive strategies [106]. It is important to understand that the mixture of the natural products associated with EV71 preventi on and control of the infection were developed through the HFMD as well as other neurological or cardiovascular symptoms.

There are some natural components to support the context of the disease control based on clinical perspectives to limit the effect of infection caused by the EV71. Hydrolyzable ellagitanninsare one of the most effective natural com ponents supporting therapeutic strategies using a hexahydroxy diphenoyl unit to control infection. It has been observed that the components of hydrolyzable ellagitannins such as geraniin, chebulagic acid, and punicalagin are responsible for providing a prolonged survival time and reduced mortality rate by inhibiting the viral replication. On the other hand, chebulagic acid application as a natural therapeutic agent is responsible for restructuring the context of viral absorption within the host cell. There is evidence for the inhibition of EV71 associated with the application of chebulagic acid, geraniin, and punicalagin.

It has been observed that alkaloids can restrict the elongation of the polyprotein of the virus during protein synthesis by interrupting the process of viral protein synthesis. However, synthetic antiviral components play an important role in preventing the infection. During infections, pleconaril is an important synthetic antiviral component in the strategy to reduce the mortality rate from the respective virus. Pleconaril, a synthetic component, is responsible for promoting the antiviral activity within the organism by restricting viral replication. Imidazolidinone is a series of derivatives associated with the Pleconaril responsible for the inhibitory action of SCARB2 and PSGL-1. Evidence from Taiwan has demonstratedthe application of imidazolidinone derivatives to control the function of the host-pathogen interaction [107].

Low cellular toxicity of some synthetic molecule peptideshas been observed when used as antivirals against EV71. Rupintrivir is an important synthetic antiviral agent which can promote the irreversible peptidomimetic inhibitors based on arestrictionprocessand inhibit viral replication [108]. Moreover, the synthetic component Rupintrivir can inhibit the action of VP1 to inhibit the host-pathogen interaction and relative responses. Sorafenib canimprovethe post-infection condition by restrictingthe viral RNA replication throughblocking the signaling pathways such as p38 and EPK [109]. Additionally, immunoglobulin treatmentssuch as IVIG havebeen demonstrated to be effective in the neutralization of antibodies. Clinical surveillance, including preventive measures againstEV71 infection, can assist in improving the understanding of epidemic conditions and health measures required in the process of disease control.

## 7. Conclusions

The study has identified various factors including the clinical, molecular, and immunological responses associated withEV71. The pathological aspects including the factors associated with the virulence of the virus has been examined. Further clarification of the epidemiological evidence at clinical and molecular levels will enable effective vaccine development as well as other sustainable preventive strategies. 

## Figures and Tables

**Figure 1 viruses-14-02190-f001:**
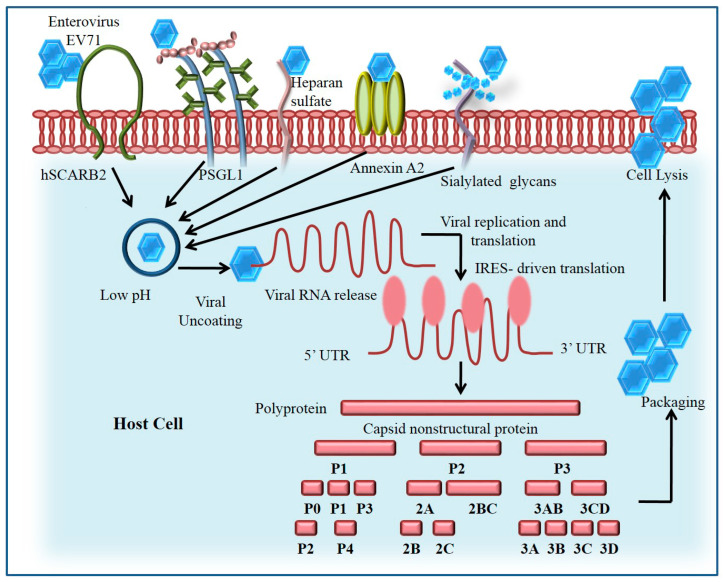
Invasion and replication of EV71 inside the host cell. EV71 attaches to cell surface receptors, co-receptors or other proteins. Endophilin binds to the cytoplasmic motif of activated receptors with SH3 domain and mediates the formation of endocytic carrier with actin polymerization and dynamin. The mechanism of immunogenic evasion of EV71 is based on the conformational changes in the antigen structure. A single polyprotein is produced by IRES-driven translation which is then cleaved into proteins by proteolysis.

**Figure 2 viruses-14-02190-f002:**
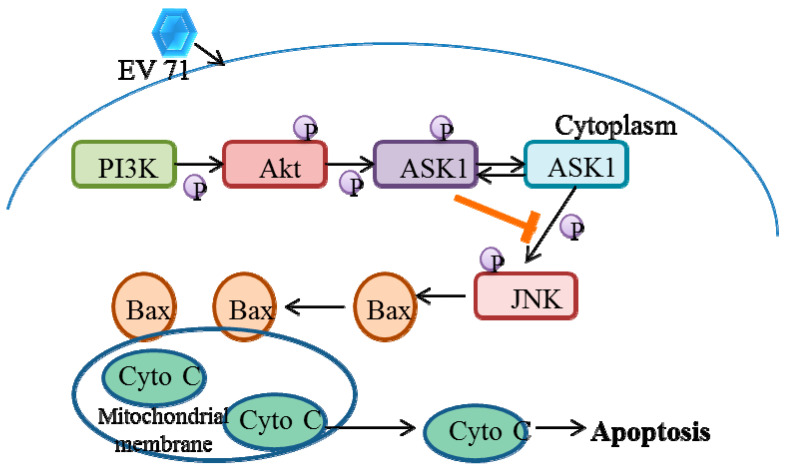
EV71-induced JNK-mediated apoptosis pathway inhibited by PI3K/Akt activation through phosphorylating and negatively regulating ASK1.

**Figure 3 viruses-14-02190-f003:**
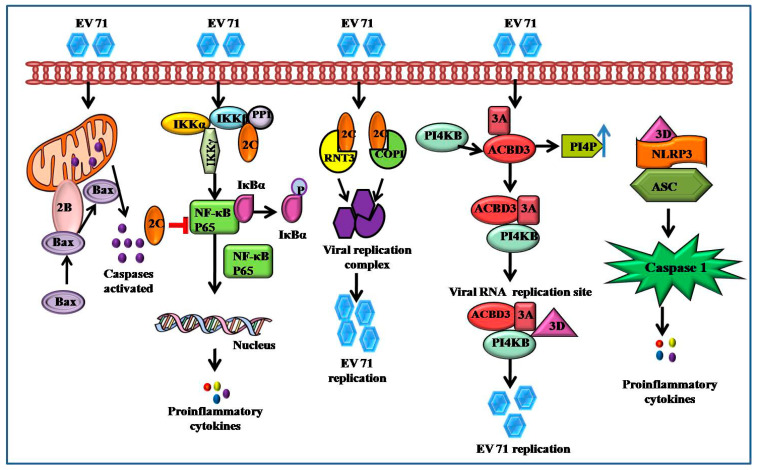
A graphical illustration of the role of each protein in EV71-induced signaling pathways. By directly interacting and activating the pro-apoptotic protein Bax, 2B triggers cell apoptosis. The overexpression of anti-apoptotic protein Bcl-XL prevents 2B-induced caspase activation and cytochrome C release. 2C protein exhibits both RNA and membrane-binding activity, interacts with reticulon 3 (RNT3), then combines with double-stranded RNA viruses to form a viral replication complex and participates in viral replication. By interacting with 3A protein, the Golgi protein acylcoenzyme A binding domain-containing 3 (ACBD3) facilitates EV71 replication, and triggers the generation of IL-1β by activating NLRP3 inflammasome.

**Figure 4 viruses-14-02190-f004:**
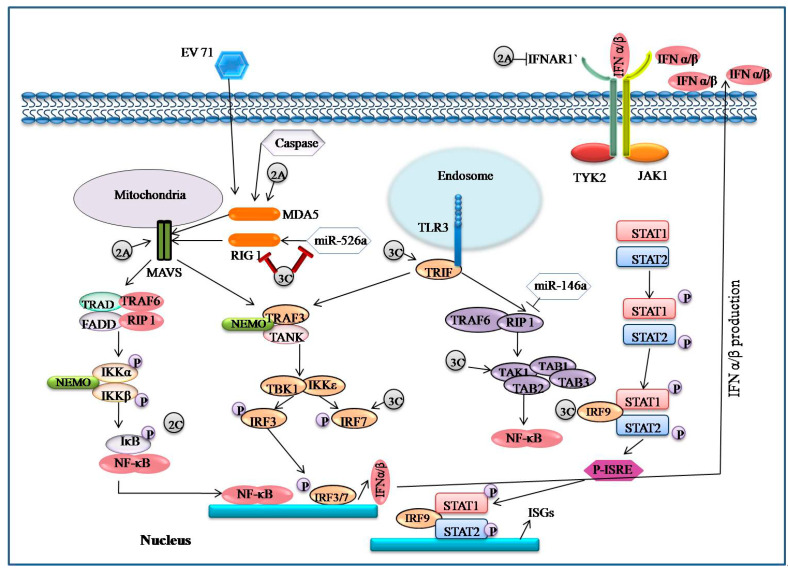
Evasion of PRRsmediated signaling pathways by EV71. By means of the non-structural proteins 2A, 2C, and 3C, EV targets the PRRs signaling pathway. 2A degrades IFNAR1 and cleaves MAVS and MDA5. RIG-1 and MAVS complex is cleavedby 3C which also inactivates TRIF. The activation of RIG-1 bymir-526a is inhibited by 3C and by promoting the expression of miR-146a.

## Data Availability

Not applicable.

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
