# Peer review of "Activation of Host Cellular Signaling and Mechanism of Enterovirus 71 Viral Proteins Associated with Hand, Foot and Mouth Disease"

_viruses, 2022, doi:10.3390/v14102190_

Round 1

Reviewer 1 Report

1. A important content/studies in the second section " molecular mechanisms associated with host-Pathogen Interaction in EV71" that  described in the Fig. 1 is missing and shall be reviewed and discussed in this review article; the mechanisms/pathways of enterovirus entry into the cells such as clathrin-,  caveolin- and others mediated endocytosis.

2. Pathogenesis study  induced by enterovirus in animal study that associated with clinical features is also missing , for example; neurotoxic diseases via. TLR/iNOS signaling shall be discussed in the content of 3.3 section.

Author Response

Response to Reviewer’s comment

The authors are really thankful and appreciate you for taking the time to analysis and share your experience with us. The comments given are very helpful and we warmly acknowledge the suggestions. The modifications have been made along with the reviewer's comments which we feel need to be corrected were made in track change mode.

Comment 1: A important content/study in the second section " molecular mechanisms associated with host-Pathogen Interaction in EV71" that is described in Fig. 1 is missing and shall be reviewed and discussed in this review article; the mechanisms/pathways of enterovirus entry into the cells such as clathrin-,  caveolin- and others mediated endocytosis.

Response to comment 1: We are grateful to the reviewers for keenly looking into the manuscript and coming up with valuable suggestions. We have modified the manuscript according to the comments in track change mode which will be easy to identify the changes made in the manuscript. We added the clathrin- and caveolin- mediated endocytosis in the second section of the manuscript i.e “Molecular mechanism associated host-pathogen interaction in EV71” with the respective references. We have made some changes in Figure-1 and Figure- 2 and their respective legends.

Comment 2: Pathogenesis study induced by enterovirus in animal study that associated with clinical features is also missing, for example; neurotoxic diseases via. TLR/iNOS signaling shall be discussed in the content of 3.3 section.

Response to comment 2: The authors apologize and are grateful to the reviewers for pointing out the mistake.  “The study of animal pathogenesis using EV71” is added in section 3.1, however, we want to acknowledge that the TLR, as well as other signaling pathways, have been discussed in the 5th section of this review article. We would like to request to the reviewer consider it. The references were arranged accordingly.

Reviewer 2 Report

 Activation of host cellular signaling and mechanism of EV-A71 pathogenicity are interesting topics. In this manuscript, the authors investigate the latest findings about the interaction of viral ligands to the host receptor as well as the activation of immune related signaling pathways for the activation of innate immunity and involvement of different cytokines and chemokines in the host pathogen interaction of EV-A71, and Interferons production/signaling pathway and their inhibitory effects. Overall, this manuscript is well written. I enjoy reading it. I endorse its publication in Viruses.

Author Response

Response to comment: On behalf of all authors we would like to thank the reviewer for the appreciation and to give valuable time to review our manuscript. We would like to continue our hard work to do better science in this domain.

Round 2

Reviewer 1 Report

No any suggestions for authors